# Exploring the Epidemiology of Cancer after Solid Organ Transplantation (EpCOT): an observational cohort study

Adnan Sharif [1,2] Javeria Peracha,[1] David Winter,[3] Raoul Reulen,[3] Mike Hawkins[3]

► Prepublication history and supplemental material for this paper is available online. To view these files, please visit the journal online (http://dx.doi.org/10.1136/bmjopen-2020-043731).

[1]Renal Medicine, University Hospitals Birmingham NHS Foundation Trust, Birmingham, UK
[2]Institute of Immunology and Immunotherapy, University of Birmingham, Birmingham, UK
[3]School of Health and Population Sciences, University of Birmingham, Birmingham, UK

**Correspondence to**
Dr Adnan Sharif;
A.Sharif@bham.ac.uk

## ABSTRACT

**Introduction** Solid organ transplant patients are counselled regarding increased risk of cancer (principally due to their need for lifelong immunosuppression) and it ranks as one of their biggest self-reported worries. Post-transplantation cancer is common, associated with increased healthcare costs and emerging as a leading cause of post-transplant mortality. However, epidemiology of cancer post-transplantation remains poorly understood, with limitations including translating data from different countries and national data being siloed across different registries and/or data warehouses.

**Methods and analysis** Study methodology for Epidemiology of Cancer after Solid Organ Transplantation involves record linkage between the UK Transplant Registry (from NHS Blood and Transplant), Hospital Episode Statistics (for secondary care episodes from NHS Digital), National Cancer Registry (from cancer registration data hosted by Public Health England) and the National Death Registry (from NHS Digital). Deterministic record linkage will be conducted by NHS Digital, with a fully anonymised linked dataset available for analysis by the research team. The study cohort will consist of up to 85 410 solid organ transplant recipients, who underwent a solid organ transplant in England between 1 January 1985 and 31 December 2015, with up-to-date outcome data.

**Ethics and dissemination** This study has been approved by the Confidentiality Advisory Group (reference: 16/CAG/0121), Research Ethics Committee (reference: 15/YH/0320) and Institutional Review Board (reference: RRK5471). The results of this study will be presented at national and international conferences, and manuscripts with results will be submitted for publication in high-impact peer-reviewed journals. The information produced will also be used to develop national evidence-based clinical guidelines to inform risk stratification to enable risk-based clinical follow-up.

**Trial registration number** NCT02991105.

## Strengths and limitations of this study

► Population-based cohort linked dataset of all solid organ transplant recipients in England to provide comprehensive coverage.
► A large, retrospective analysis using national linked datasets of solid organ transplant recipients at risk for developing cancer.
► Robust and collaborative data linkage of routinely collected administrative data, for the purpose of exploring important clinical outcomes of research interest post-transplantation.
► Missing variables and confounding data may limit interpretation of the analysis.
► Data are England-specific and may not be translatable to other cohorts.

## INTRODUCTION

Solid organ transplantation is established therapy for patients with end-stage organ dysfunction, providing life-saving (eg, heart, lung and liver) or life-enhancing (eg, kidney and pancreas) treatment. However, cancer is now emerging as one of the leading complications after solid organ transplantation. Transplant recipients have an increased risk of cancer compared with that expected from the general population[1–3] and comparable risk to immunodeficiency states such as HIV/AIDS,[4] with similar preponderance for cancers with a viral component.[5] Multifactorial aetiology underlies cancer risk after solid organ transplantation, but immunosuppression, compulsory to prevent allograft rejection, contributes significantly to cancer risk.[6 7] Standardised incidence ratio for development of any post-transplantation cancers (excluding non-melanoma skin cancers) remain constantly increased starting from 2 years post-transplantation in the UK,[3] meaning the relative excess risk of cancer increases with time. Transplant recipients are achieving better long-term survival[8] and, among kidney transplant patients with more than 20 years graft function, de novo cancer develops in 37% of patients.[9] This is likely due to improved post-transplant longevity leading to increased cumulative exposure to cancer risks such as immunosuppression. In summary, post-transplantation cancer is now the leading cause of death after kidney transplantation in the UK;[10] death from cancer post-transplantation is in excess of that expected from the general population;[11–14] it ranks as one of the biggest worries

BMJ

for post-transplant patients,[15] and development of post-transplantation cancer is associated with increased healthcare cost.[16]

Some reports on cancer epidemiology after solid organ transplantation focus on historical eras and practice has changed over the last decade. Immunosuppression is significantly different, especially since 2006 when dissemination of the SYMPHONY study results led to switches from ciclosporin to tacrolimus-based immunosuppression.[17] This is important as tacrolimus has been shown in liver transplant recipients to increase risk of post-transplantation cancer compared with ciclosporin,[18] with an observed dose–effect relationship between tacrolimus and post-transplantation cancer.[19] Cancer risk appears decreased with sirolimus after kidney transplantation[20 21] and increased with azathioprine after solid organ transplantation,[22–24] but some studies (limited to kidney transplantation) suggest no difference in cancer risk between immunosuppressant agents.[25] Induction immunosuppression is also more commonly used; while T-cell depletion induction has been linked with increased risk of post-transplantation cancer,[26] induction therapy can lead to reduced exposure to maintenance immunosuppression, further confounding cancer risk assessment.[27] Evolving immunosuppression, changing transplant practice and selection of solid organ donors/recipients with different characteristics justify a more contemporary and time-dependent analysis of post-transplantation cancer.

Data related to the epidemiology of post-transplantation cancer predominantly comes from outside the UK (eg, the USA), but this may not be directly translatable. Immunosuppression use, the biggest modifiable risk factor for post-transplantation cancer,[28] differs between the UAS and the UK. For example, T-cell depletion therapy is the predominant induction agent in the USA versus non T-cell depletion (eg, basiliximab) in the UK.[29] Cumulative exposure to immunosuppression will be different, with financial coverage limitations in the USA extending to immunosuppression provision[30] and these factors could possibly contribute to inferior post-transplant outcomes observed in the USA compared with Europe.[31 32] Kidney allograft recipients in England have more pretransplant cancer history, more post-transplant cancer occurrence but superior overall graft survival and all-cause mortality when compared with contemporaneous recipients in New York State,[33] suggesting post-transplantation cancer risk (and outcomes) is not comparable. Study cohorts from Australia and/or New Zealand are more aligned to UK practice from an immunosuppression perspective. However, they have different demographics (eg, less minority ethnic recipients) and burden of medical comorbidities (eg, cause of kidney failure) that limits direct translation of outcomes.[34] With only two epidemiology studies focused on post-transplantation cancer in the UK,[3 13] further studies in the UK are urgently warranted to inform patients and professionals.

Absence of relevant epidemiological data impacts on patient counselling regarding increased risk of post-transplantation cancer, which is accepted by transplant candidates as a recognised complication from immuno-suppression. However, individual risk stratification is not possible and this has implications in the contemporary era of consent.[35] Personalised counselling is difficult as we cannot predict individual post-transplant cancer risk and screening is only available for a minority of cancers. Gaining a clearer understanding of post-transplantation cancer risk, morbidity and mortality for specific transplant groups/cohorts will aid transplant discussions. This is especially important for cancers of greater incidence and risk of adverse outcomes after solid organ transplantation (eg, post-transplant lymphoproliferative disorders). Kidney Disease Improving Global Outcomes guidelines for the care of (kidney) transplant patients report very poor level evidence when it comes to cancer, with either level 2C or non-graded evidence.[36] This translates into lack of consensus on cancer screening advice post-transplantation,[37] lack of evidence-based guidance on how best to manage development of post-transplant cancer[38] and overall poor outcomes post-transplantation if transplant recipients develop cancer.[11–13]

To understand the epidemiology of post-transplant cancer in the UK, it is imperative to bring together multiple sources of information relating to the same solid organ transplant recipient who may develop cancer contained across different registries or datasets. Big data linkage, the process of bringing together and linking large volumes of healthcare records, can help identify factors and associations that would otherwise be difficult to robustly determine. The ability to link data can significantly increase the value that can be derived from individual datasets—which are often collected for specific regulatory purposes with considerable effort and expense—and maximise research benefit for patients. To that effect, Epidemiology of Cancer after Solid Organ Transplantation (EpCOT) has the aim of exploring prespecified research questions of importance to the solid organ transplant community using big data record linkage of national healthcare datasets. The study protocol is registered with clinicaltrials.org, and study results will be reported in accordance with

| Table 1 | Snapshot of the UK Transplant Registry cohort | | | |
|---|---|---|---|---|
| | 1 January 1985–31 December 1994 | 1 January 1995–1 December 2005 | 1 January 2006–31 December 2015 | Total |
| Kidney | 14341 | 16172 | 23436 | 53949 |
| Pancreas | 76 | 451 | 1835 | 2362 |
| Heart | 3271 | 2533 | 1457 | 7261 |
| Lung | 1160 | 1593 | 1683 | 4436 |
| Liver | 3296 | 6846 | 6925 | 17067 |
| Pancreas islets | 0 | 0 | 120 | 120 |
| Abdominal/intestinal | 4 | 47 | 164 | 215 |
| Total | 22148 | 27642 | 35620 | 85410 |

Strengthening the Reporting of Observational Studies in Epidemiology (STROBE) guidelines.[39]

## METHODS AND ANALYSIS

This epidemiological study will undertake record linkage between national datasets to link up the pathway from solid organ transplantation and subsequent risk of developing cancer. Deterministic record linkage will be undertaken by NHS Digital using identifiable patient data (specifically NHS number, date of birth, sex and postcode) between the following datasets:

1. UK Transplant Registry (UKTR). The UKTR is maintained by NHS Blood and Transplant (NHSBT) and holds data for all solid organ transplants performed in the UK, with data submission mandatory for transplant centres. UKTR will provide data for the central study cohort of 85 410 patients who received a solid organ transplant between first January 1985 and 31st December 2015 in England, with donor-, recipient- and transplant-specific variables. An overview of the cohort is provided in table 1, with key demographics shown in table 2 and the data dictionary in online supplemental file 1. The final cohort for analysis will be determined after excluding any repeat transplant recipients (only the first transplant will be considered in the analysis), non-English residents who received their transplant in England and anyone who refused consent for their data to be used.

2. Civil Registration—Deaths (NHS Digital). National death registration data are held by NHS Digital. For those patients who have died, the date and underlying cause of death from death certificate codes will be obtained for any death occurring in the UK (any deaths occurring outside of the UK will be missed, but date of emigration (loss to follow-up) will be provided). For date of death, only a flag indicating that death has occurred and the months from transplantation to death

| Table 2 | Demographics of UK Transplant Registry cohort | | | | | | | |
|---|---|---|---|---|---|---|---|---|
| | **Kidney (n=53 949)** | **Pancreas (n=2362)** | **Heart (n=7261)** | **Lung (n=4436)** | **Liver (n=17 067)** | **Pancreas islets (n=120)** | **Abdominal/ Intestinal (n=215)** | **Missing or unreported data** |
| First graft | 46 347 | 2230 | 7115 | 4305 | 15 241 | 74 | 201 | 0 |
| Part of multiorgan transplant | 2199 | 1994 | 1136 | 1027 | 441 | 0 | 137 | 0 |
| Donor | | | | | | | | |
| Living | 12 475 | 0 | 0 | 24* | 380 | 0 | 0 | * |
| DBD | 35 310 | 2039 | 7246 | 4199 | 15 551 | 110 | 215 | |
| DCD | 6164 | 323 | 14 | 224 | 1136 | 10 | 0 | |
| Female recipient | 20 510 | 1006 | 2008 | 2062 | 7517 | 87 | 93 | 48 |
| Ethnicity of recipient | | | | | | | | |
| White | 33 556 | 2060 | 3963 | 3237 | 12 073 | 117 | 181 | 19 029 |
| Asian | 4641 | 115 | 269 | 75 | 1476 | * | 15 | |
| Black | 2369 | 69 | 77 | 26 | 438 | 2 | 5 | |
| Chinese | 300 | 5 | 11 | 2 | 107 | 0 | 0 | |
| Mixed | 122 | 5 | 15 | * | 23 | 0 | * | |
| Other | 545 | 19 | 39 | 9 | 408 | 0 | 3 | |
| Recipient age (years) | | | | | | | | |
| 0–9 | 1329 | 37 | 611 | 86 | 2028 | 0 | 98 | 22 |
| 10–19 | 2860 | 20 | 733 | 392 | 946 | 0 | 15 | |
| 20–29 | 6546 | 168 | 746 | 734 | 1176 | 0 | 21 | |
| 30–39 | 9719 | 781 | 813 | 607 | 1745 | 10 | 23 | |
| 40–49 | 12 285 | 912 | 1556 | 864 | 3390 | 42 | 26 | |
| 50–59 | 11 972 | 397 | 2265 | 1288 | 4812 | 48 | 20 | |
| 60–69 | 7733 | 47 | 531 | 464 | 2861 | 17 | 11 | |
| 70–79 | 1464 | 0 | 3 | 1 | 106 | 3 | 1 | |
| 80+ | 25 | 0 | 0 | 0 | 0 | 0 | 0 | |

*Partial solitary lung.
DBD, donation after brain death; DCD, donation after cardiac death.

(to calculate time interval) will be processed in line with legal approval.

3. National Cancer Registry: National Cancer Registration and Analysis Service (NCRAS) from Public Health England (PHE). NCRAS collects information about every patient diagnosed with cancer in England. All variables of cancer-related interest will be linked to our cohort (between 1 January 1986 and 31 December 2017 unless stated otherwise) and include cancer site (C00c-C097x and D00-D48), histology, treatment, chemotherapy (between 1 April 2012 and 31 March 2018) and radiotherapy (between 1 April 2009 and 31 March 2019). The data dictionary is provided in online supplemental file 2.

4. Hospital Episode Statistics (HES). HES is an administrative dataset collating information on all secondary care events in England, held by NHS Digital. From HES we can determine adverse health outcomes or procedures requiring hospitalisation (eg, inpatient, outpatient and accident and emergency attendance), for which the risk may be increased as a result of having had a solid organ transplant. Period at risk of adverse health outcomes would be from 1997 until the most recent date for which HES is known to be complete. The data dictionary is provided in online supplemental file 3.

### Record linkage methodology

Data linkage methodology will involve UKTR and NCRAS sending to NHS Digital only a list of identifying details for their respective cohorts. NHS Digital will link the UKTR and NCRAS cohorts, determining which patients are common in both datasets. NHS Digital will then provide a list of pseudonymous IDs of patients common

in both datasets to UKTR and NCRAS, respectively. This will facilitate extraction of clinical/health data from their datasets, respectively, for provision to the University of Birmingham identifiable by the pseudonymous ID only. NHS Digital will facilitate linkage to HES and 'Civil Registration—Deaths' to extract records and supply anonymised records to University of Birmingham with the common pseudonymous ID. The University of Birmingham will then facilitate linkage between the NHS Digital, UKTR and NCRAS datasets using the pseudonymous ID common to all datasets. See figure 1 for a summary of data flows for the EpCOT study.

### Data security and governance

There will be no need, requirement or possibility to reidentify individuals after record linkage by the University of Birmingham (acting as data processor). Received data will be stored on a separate array of disks and accessed on a mapped drive as part of the University of Birmingham campus network. Access to this mapped drive will be limited to the named university researchers who can only access the University of Birmingham campus network with a personalised username/password combination. The personal computers used to access the data will have additional virus checking/data security software installed (Malwarebytes). The separate array of disks will reside in a controlled environment with access only by accredited university IT staff. The data from this disk array will be backed up under a separate storage policy, which once deleted will make the data unavailable for restore.

The linkage, processing and analysis of the data from UKTR, NCRAS and NHS Digital will only be carried out the specified researchers within the University of Birmingham as the data processor who are appropriately

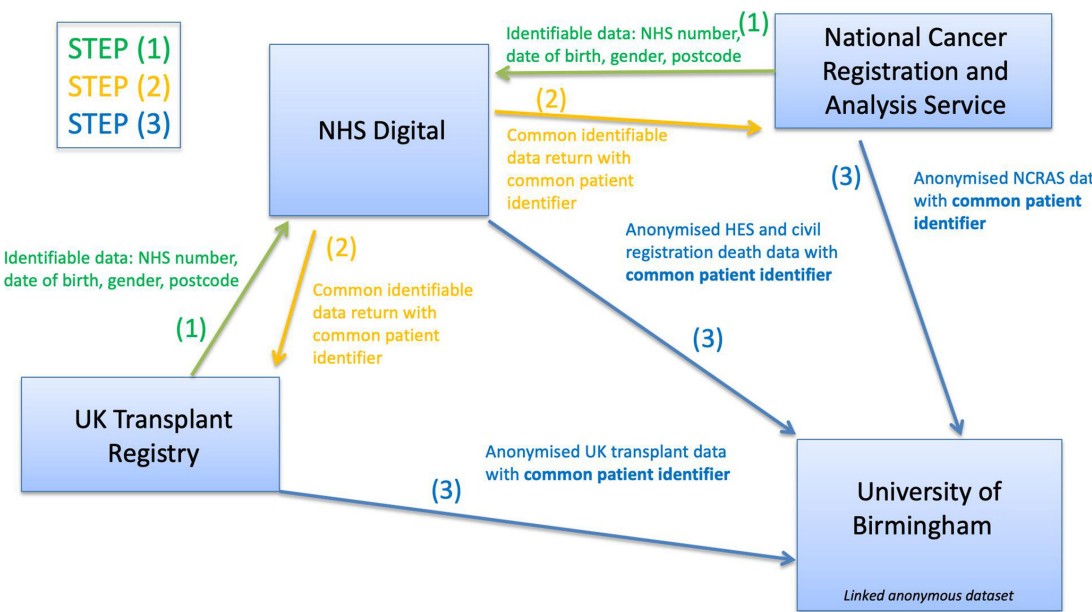

**Figure 1** Flow of study data for the Epidemiology of Cancer After Solid Organ Transplantation project between data providers, NHS Digital and the University of Birmingham as data processor. HES, Hospital Episode Statistics; NCRAS, National Cancer Registration and Analysis Service.

trained in data protection and confidentiality. There will be no attempt made to link any data requested under this application to any held under other Data Sharing Agreements. This database of the pseudonymised data will be accessible only to named personnel within the project and will be kept in auditable documents. In addition, analysts fulfilling database administrator role (who are substantive members of the University of Birmingham) will also have access to the database but undertake no processing. At the end of the project, the data will be destroyed in line with strict policies and procedures on the destruction of data. No record level information will leave the data processor or aggregated data without small number suppression in line with the NHS Digital HES Analysis Guide.

## Prespecified research questions

1. To compare observed and expected risks of specific causes of deaths, in particular cancer-related death, by linking the UKTR with the National Death Registry to obtain underlying causes of death and to determine factors related to increased risk of specific causes of death post-transplantation. General population mortality rates will be used to calculate the expected number of deaths from specific causes and identify subgroups of post-transplant patients (eg, age, sex, ethnicity and organ type) at excess risk compared with expected risk.
2. Investigate survival and causes of death after cancer in post-transplant patients versus individuals from the general population with a similar de novo cancer of the same age, sex and calendar year of diagnosis.
3. Compare observed and expected risks of specific cancer types post-transplantation by linking the UKTR with the National Cancer Registry to obtain observed numbers of cancers and to determine factors related to increased risk of specific types of cancer. General population cancer incidence rates will be used to calculate expected numbers of cancers of specific type and to identify subgroups of post-transplant patients at excess risk of specific cancers compared with expected.
4. Estimate risk of morbidity requiring hospitalisation both generally and that associated with development of post-transplantation cancer by linking the UKTR with HES. Risk of hospital admissions and procedures (eg, surgery) for specific morbidities will be investigated. We will calculate expected risks for specific conditions requiring hospitalisation, enabling identification of specific subgroups of post-transplant patients at excess risk compared with expected.

## Detailed research plan

### Compare observed and expected risks of specific causes of death

Each of the patients in the UKTR enters risk at the date of transplantation and contributes person-years until the exit date. Standardised mortality ratios (SMRs) and absolute excess risks (AERs) will be calculated as O/E and $[(O–E)/py]\times1000$, where 'O' and 'E' are the observed and expected numbers of deaths, respectively, and 'py'

is the total person-years at risk. To examine variation in SMRs or AERs across factors (eg, age, sex, ethnicity, organ type, period of follow-up and calendar year of diagnosis), multivariable Poisson regression models will be used. Cumulative incidence of deaths from specific causes, in particular cancer, will be estimated (treating other causes of death as competing risks). All statistical analyses will use STATA V.16 unless otherwise stated.

### Investigate survival and causes of death after cancer in transplant patients

Comparators will be individuals from the general population with a similar de novo cancer of the same age, sex and calendar year of diagnosis. On one hand, heightened surveillance among solid organ transplant survivors should detect any cancers at an early stage and, therefore, reduce cancer-related mortality. On the other hand, previous treatment with immunosuppressants may limit treatment options for cancer and hence increase cancer-specific mortality. Survival comparisons will use Cox regression. We shall compare all-cause and cause-specific mortality of patients who have undergone a solid organ transplant and developed a cancer with that of individuals who developed a similar de novo cancer using Cox regression and/or Poisson regression analysis. Summary data of patients from the general population who developed cancer, without any transplant exposure after record linkage by NHS Digital, will be obtained from NCRAS.[40]

### Compare observed and expected risks of specific cancer types

Data from NCRAS will identify diagnosis of post-transplantation cancers in terms of date and site/type. The initial analysis will involve constructing a table with rows corresponding to variables relating to characteristics (eg, age, sex and organ) of all transplant patients and the columns corresponding to the cancer site/types. Observed and expected numbers in each cell of this table will identify any evidence of an excess. Poisson regression as described earlier will be used to determine subgroups of patients at a substantially increased risk of specific cancers.

We will also compare cancer risk for transplant patients treated before/after 2006 (introduction of tacrolimus-based regimens) to evaluate whether more recently treated patients have increased/decreased post-transplantation cancer risk.

### Estimate risk of morbidity requiring hospitalisation

Two types of analyses will be possible using UKTR-HES linked data:

1. Internal analyses: risk of specific adverse outcomes post-transplantation will be compared over the period at risk using Poisson regression, to identify particular subgroups (eg, in terms of attained age, sex, type of transplant, age at diagnosis, period of follow-up and calendar year of diagnosis) at greatest risk of the specific adverse health outcomes.

2. External analyses: obtaining general population HES events classified by age, sex and calendar year and dividing by appropriate general population estimates will enable external comparisons. We will use Poisson regression to compare risk of adverse health outcomes, over the period at risk, between transplant patients and the general population adjusting for attained age, sex and calendar year in the model. Patient subgroups (defined in terms of attained age, sex, type of transplant, age at diagnosis, period of follow-up and calendar year of diagnosis) at substantial excess risk compared with expected will thus be identified.

Complementary to our analyses specified previously, a data-driven methodology incorporating latest developments in artificial intelligence (eg, machine learning) will be explored, enabling us to rank the importance of predictors for post-transplantation cancer if possible. By developing and applying models to approximate complex and yet-unknown interactions between clinical features, better prediction of patient outcomes could inform certain streams of post-transplantation care.

## Analytical considerations

### Adequate power for study events

In previous work by Collett *et al*,[3] 37 617 solid organ transplant recipients in the UK were analysed, while in the current proposal, we included more than double (n=85 410, although the final cohort after exclusions still to be determined). Using a conservative approach, assuming the number of expected cancers will be twice that from previous work by Collett and colleagues (although expected number will likely be much greater as the cohort has aged), then we will be able to detect Standardised Incidence Ratio (SIRs), of magnitudes as shown in table 3, with 80% power when using a 5% significance level.

Thus, apart from cancer sites of the anus, lip and Kaposi sarcoma, which are the the most commonly occurring, we should be able to detect an SIR of 2.0 or less with at least 80% statistical power. Similar logic would apply to our analyses relating to hospitalisations and mortality.

### Missing data

Missing data are a reality of both administrative and registry data. Examples of the degree of missing data from the UKTR have been indicated in table 2, but the final amount after record linkage is undertaken will not be known and in some instances could be reduced (eg, missing ethnicity data in UKTR may be available from HES or NCRAS which also contain this data field).

Our approach for dealing with missing data will be to only analyse the available data (ie, ignoring the missing data) and to conduct sensitivity analyses to explore whether missing data could have biased our results; or, where appropriate, to impute missing data and account for the fact that missing values were imputed with uncertainty (eg, multiple imputation).

**Table 3** Adequate power for EpCOT study events

| Site of cancer development | Estimated expected number of cancers | Minimal SIR that can be detected |
|---|---|---|
| All cancers excluding non-melanoma | 1650 | 1.1 |
| Skin: non-melanoma | 294 | 1.1 |
| Lung and bronchus | 252 | 1.1 |
| Breast | 243 | 1.1 |
| Colorectal | 200 | 1.2 |
| Prostate | 197 | 1.2 |
| Bladder | 64 | 1.3 |
| Non-Hodgkin's lymphoma | 56 | 1.3 |
| Skin: malignant | 53 | 1.4 |
| Stomach | 49 | 1.4 |
| Oesophagus | 44 | 1.4 |
| Kidney | 39 | 1.5 |
| Pancreas | 38 | 1.5 |
| Leukaemia | 37 | 1.5 |
| Ovary | 35 | 1.5 |
| Uterus | 31 | 1.5 |
| Multiple myeloma | 21 | 1.7 |
| Cervix | 18 | 1.8 |
| Oral cavity | 16 | 1.8 |
| Liver | 16 | 1.8 |
| Hodgkin's lymphoma | 10 | 1.9 |
| Thyroid | 9 | 2.0 |
| Anus | 5 | 3.0 |
| Lip | 2 | 4.0 |
| Kaposi sarcoma | 1.4 | 5.5 |

EpCOT, Epidemiology of Cancer after Solid Organ Transplantation; SIR, Standardised Incidence Ratio.

### Quality assurance for submitted data and record linkage

Record linkage will be facilitated by NHS Digital. The research team is therefore reliant on robust quality assurance mechanisms from both data senders and NHS Digital Data Linkage and Extract Services to ensure a robust and comprehensive record linkage process. We have worked very closely with all partners, carefully selecting fields from the available data dictionaries and checking their suitability and viability for record linkage. Cost recovery frameworks have been factored into our budget to ensure robust data extraction and quality assurance processes are in place. NHS Digital charges also includes list cleaning, a validation process using demographic data to ensure accuracy and to improve linkage outcomes. A record linkage report will be available to check record linkage success rate.

## Patient and public involvement (PPI)

Patient involvement has been critical in the design of our research aims to ensure our research questions are both relevant and appropriate. We have developed our research aims after discussion with key stakeholders like patients to ensure we are seeking answers to questions that they wish to be answered and further study into post-transplantation cancer meets those needs. PPI group meetings involving transplant candidates and transplant recipients, hosted as part of the Trust's initiative to ensure adequate patient and public representation in research drives, have confirmed qualitative study findings that post-transplantation cancer is a leading concern and an area where preventative strategies would be overwhelmingly welcomed. Patients were very receptive to our strategy of record linkage, both using anonymised national data resources and also using linked primary/secondary care data after informed consent if required for any future extension work.

## ETHICS AND DISSEMINATION
### Approvals for record linkage

The EpCOT study research plan has been reviewed and approved by our data contributing partners: NHS Digital, the UKTR and PHE. Their support, collaboration and advice have been critical for the development of data flows, and we plan to work in close partnership at various stages of the project. Our wide consortium of partners has demonstrated their strong support to ensure the success of this project over many years of development.

The EpCOT project has attained institutional (*RRK5471*) and ethical (*15/YH/0320*) approval. The project also has section 251 approval from the Confidentiality Advisory Group (*16/CAG/0121*), under Regulation 5 of the Health Service (Control of Patient Information) Regulations 2002, for the record linkage to proceed by NHS Digital.

### Dissemination plan

The aim of this project was to produce targeted studies looking at the epidemiology of post-transplantation cancer, and these data will be disseminated through presentations at national/international congresses and submitted for publication in leading medical journals. The aim was also to develop standards of care guidelines, working with the Standards Committee of the British Transplantation Society to provide evidence-based clinical evidence for direct patient benefit and counselling. These will be made freely available to the community at their website (https://bts.org.uk/guidelines-standards/). It is also the aim to plan patient-focused dissemination in plain English through various channels of communication for patients and the general public.

Outputs will only contain aggregate data with small numbers suppressed in line with the NHS Digital HES Analysis Guide.[41] Both NHSBT and PHE (which will provide the initial data cohorts on this project) have

mechanisms for dissemination of information to both professional and general populations, and we will use these to share the results from EpCOT widely. We also aimed to do similar research presentation at Research Open Days at both University Hospitals Birmingham and the University of Birmingham. Summary research findings are actively disseminated via social media channels (eg, @AdnanSharif1979, @UHBResearch and @ImmunologyUoB) while tagging our collaborative partners (@NHSBT_RD, @PHE_uk and @NHSDigital). Project partners will be encouraged to disseminate their involvement through all social media channels.

## DISCUSSION

Large epidemiological studies from the USA (using data from the Transplant Cancer Match Study) have recently reported post-transplant cancer epidemiology outcomes for a cohort of 221 962 solid organ transplant recipients, of whom 15 012 developed cancer.[42] One of the main findings from Noone and colleagues is cancer-related mortality increases steadily with post-transplantation time duration, reaching 15.7% of deaths (810 per 100 000 person-years) after 10 or more years after transplantation. While a large cohort, one of the limitations of this dataset is the incomplete national coverage; a total of 17 cancer registries provided data regarding incident cancers (excluding nonmelanoma skin cancer) that covers approximately one-half of the US transplant population with varying years of coverage data. To the best of our knowledge, EpCOT is one of the largest population-cohort studies exploring cancer epidemiology after solid organ transplantation. Using record linkage of national datasets with extended years of coverage, this project will provide detailed information for solid organ transplant recipients with cancer-related coverage at a countrywide level.

In order to fulfil its regulatory mandate, NHSBT collects data for all solid organ transplant recipients including long-term follow-up information. The collection of national data enables NHSBT to fulfil its statutory obligations with regard to the effective use of organs, equitable organ offering and performance monitoring of transplant centres in terms of patient and graft outcomes. Such data have tremendous analytical potential and is available for research purposes with appropriate approval. However, the exploration of certain outcomes like cancer is limited as this information is not routinely collected by NHSBT. To overcome this issue, record linkage between the transplant registry and national cancer data is essential. The methodology of EpCOT facilitates this, with the inclusion of all secondary care episodes and death registrations, to obtain a robust, comprehensive and informative population-cohort study of post-transplantation cancer. From an epidemiological perspective, this allows detailed analyses to assess crucial estimates of the impact of cancer for solid organ transplant recipients and provides

coverage of the full patient journey from transplantation to cancer-related outcomes.

In summary, cancer is a leading cause of morbidity and mortality after solid organ transplantation. With incidence of cancer projected to increase, improving our understanding of post-transplant cancer epidemiology is essential to improve long-term patient outcomes. EpCOT will be one of the largest record linkages of solid organ transplant recipients, with contemporary and comprehensive national cancer-related outcomes, to provide answers to questions of importance to both transplant professionals and organ recipients respectively.

**Contributors** AS: substantial contributions to the conception, design of the work, seeking for funding, acquisition, analysis, interpretation of data, drafting the work, revising it critically and final approval of the version published. JP: substantial contributions to the design of the work, analysis of data, interpretation of data, critical revision and final approval of the version published. DW: substantial contributions to the design of the work, acquisition of data, storage of data, critical revision and final approval of the version published. RR: substantial contributions to the conception, design of the work, acquisition, analysis, interpretation of data, revising it critically and final approval of the version published. MH: substantial contributions to the conception, design of the work, acquisition, analysis, interpretation of data, revising it critically and final approval of the version published.

**Funding** This work was supported by a Wellcome Trust Institutional Strategic Support Fund: Digital Health Pilot Grant (419195-419186-42231786) and the Kildare Trust (no award number applicable).

**Competing interests** None declared.

**Patient consent for publication** Not required.

**Provenance and peer review** Not commissioned; externally peer reviewed.

**ORCID iD**
Adnan Sharif http://orcid.org/0000-0002-7586-9136

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
