## [Reviewer comments · BMJ Open]

ARTICLE DETAILS

TITLE (PROVISIONAL)	Protocol: observational cohort study exploring the Epidemiology of Cancer after solid Organ Transplantation (EpCOT)
AUTHORS	Sharif, Adnan; Peracha, Javeria; Winter, David; Reulen, Raoul; Hawkins, Mike

VERSION 1 – REVIEW

REVIEWER	Margaret Madeleine Fred Hutchinson Cancer Research Center USA
REVIEW RETURNED	24-Sep-2020

GENERAL COMMENTS	This manuscript describes the protocol for an important linkage study in the UK, that will allow a large and comprehensive assessment of SOT cancer outcomes over time. Future comparison with other similar large datasets internationally will allow potentially larger pooled or meta analysis in the future. Some suggestions to improve the manuscript are listed here: Page 5, Lines 25-29: “Standardised incidence ratio for development of all post-transplantation cancers (excluding non-melanoma skin cancers) remain constant...up to 15 years” does not capture the cited reference, which suggests that there is a constantly increased SIR starting 2 years post-transplant. The latter part of the sentence says (line 33) “absolute excess risk of cancer increases with time.” However, the cited reference is for a relative excess risk. This entire sentence needs to be re-phrased to more carefully reflect the cited paper. Page 6, lines 8, 9: It is not clear that tac increases all types of cancer risk in all types of cancer; references 18 and 19 are based on liver transplant, and sirolimus is reported for kidney transplants (refs 20 and 21). This summary of the important changes in treatment would be improved with reference to the organ transplanted. Page 6, line 16: induction immunosuppression is also more common – during what time period? Does impact of induction change with type of maintenance therapy? Page 6, line 21: “selection of marginal donors recipients” – not clear what authors are referring to with this phrase Line 41: Seems like a missed opportunity to forego mentioning that not all cancers have screening tests available; Also missing, mention of PTLT and its early importance
---

	Table 1: how are recipient organs (same sites and combine, such as heart and lung) counted in this table? Table 2: importance of missing race should be discussed perhaps in "Missing Data" section; could another dataset be merged that is more likely to have race/ethnicity available to add to the linkage? Page 11: Nice list of pre-specified questions. Is it possible to look at individual level pharmaceutical data as part of this linkage? If so, looking more closely at type of immunosuppression over time, in a time-dependent survival analysis by transplanted organ, would be important. Is that possible? Strongly suggest adding as a pre-specified question. Page 13, line 50: will general population data from the cancer registry be frequency matched to transplant recipients by age, gender, cancer diagnosis year?
--	--

REVIEWER	Dr Donal Sexton Trinity Health Kidney Center, School of Medicine, Trinity College Dublin, Ireland.
REVIEW RETURNED	30-Sep-2020

GENERAL COMMENTS	Thank you for the opportunity to review this study proposal. I commend the authors on a comprehensive proposal. This is an ambitious project drawing together multiple national data streams. For completion I believe it should be acknowledged by the authors that this study is not fully prospective since much of the data was collected prior to study initiation and was not specifically captured for this purpose but for other purposes. 1. Strengths and limitations section Would the authors mind explaining why only England and not the UK? perhaps related more to the cancer registry data than the NHS data? Would the authors mind citing proof of that statement that this is the "The largest national linked dataset of solid organ transplant recipients at risk for developing cancer available internationally"? i.e. the search terms in pubmed etc so that it could be replicated It may be worthwhile mentioning here also that the study is not a formal prospective longitudinal study with defined intervals of outcome ascertainment and the data was not captured specifically for this purpose, rather the investigators are proposing to use existing data. 2. Introduction section I think it's worth acknowledging that at least some of this excess cancer risk is related to improved overall survival of both recipients and the allografts over time leading to increased time at risk. difficult to justify using the word "most publications" here, there are published studies which used similar methodology to the authors proposed design. These may not have been in the UK but nonetheless used similar design.
--

	"contributes to inferior post transplant outcomes" I think it would be more appropriate to say "these factors could possibly contribute to inferior..." rather than such a definitive statement since we don't have definitive evidence that this is the case, only suggestions. With regard to big data approach - in this regard what additional or novel parameters are being collected as part of this study that would contribute to such data exploration? Otherwise if there are no new features or parameters being collected, it is unlikely that a big data approach to this data will glean any novel insights. 3. Methods section national cancer registry "all variables of cancer related interest - what is the list of these variables? could they be listed as a supplementary file? The data linkage plan and security are comprehensive. the pre-specified research questions are appropriate. is the comparison to the general population useful? Other studies have found an elevated risk of certain cancers post transplant, I would imagine a similar pattern may be observed in this study albeit with a more precise estimate of risk given the large number of participants going to be included in this study. One must also acknowledge that screening of potential recipients prior to transplant probably introduces confounding such as PSA screening for prostate cancer etc. Is there a standard UK approach to cancer work up pre-transplant? 4. compare observed and expected risks of specific cancer types section This may be difficult, although it may be obvious whether or not a period effect exists, adjusting for improvements in general medical care as well other factors such as ascertainment bias. Has a better appreciation for detection of cancer post transplant occurred over time? I would imagine this is the case at least for skin cancers. Is race well captured in each of the datasets? Can the authors adequately adjust for race? The application of a machine learning approach is limited in my opinion. The authors have mentioned to date only standard parameters such as age, sex, year of transplant. it is unlikely that machine learning techniques will produce results different from standard traditional regression approaches in this context unless there is a wide range of other parameters planned on being incorporated. 5. adequate power for study events section - Could the authors explain why so for anus, lip and Kaposi. I commend the approach to missing data. The formal involvement of PPI is commendable. Dissemination plan section The difficulty with the suggestion that guidelines for post transplant cancer surveillance will develop out of this project is that it may not, even with SIRs and excess risk assessments, unless there
--	---

	are particular inflection points after which incidence increases for different types of cancers. Tables and content are appropriate.
--	--

VERSION 1 – AUTHOR RESPONSE

Reviewer: 1

Comments to the Author

This manuscript describes the protocol for an important linkage study in the UK, that will allow a large and comprehensive assessment of SOT cancer outcomes over time. Future comparison with other similar large datasets internationally will allow potentially larger pooled or meta analysis in the future. Some suggestions to improve the manuscript are listed here:

Page 5, Lines 25-29: “Standardised incidence ratio for development of all post-transplantation cancers (excluding non-melanoma skin cancers) remain constant...up to 15 years” does not capture the cited reference, which suggests that there is a constantly increased SIR starting 2 years post-transplant. The latter part of the sentence says (line 33) “absolute excess risk of cancer increases with time.” However, the cited reference is for a relative excess risk. This entire sentence needs to be re-phrased to more carefully reflect the cited paper.

This sentence has been edited to reflect the cited work more accurately (page 4).

Page 6, lines 8, 9: It is not clear that tac increases all types of cancer risk in all types of cancer; references 18 and 19 are based on liver transplant, and sirolimus is reported for kidney transplants (refs 20 and 21). This summary of the important changes in treatment would be improved with reference to the organ transplanted.

This has now been added to page 5.

Page 6, line 16: induction immunosuppression is also more common – during what time period? Does impact of induction change with type of maintenance therapy?

Thymoglobulin, first available in Europe in 1984 and the United States in 1999). Humanized anti-CD52 mAb alemtuzumab (Campath-1H; gradually introduced in kidney transplantation in the 1990s and into the last decade), and mAbs directed against the IL-2 receptor (IL-2ra; including basiliximab, which was approved for kidney transplantation in 1998). The use of induction therapy can be used as a strategy to reduce long-term exposure to maintenance immunosuppression. This is now mentioned on page 5 with an additional citation; Wiseman AC. Induction Therapy in Renal Transplantation: Why? What Agent? What Dose? We May Never Know. *Clin J Am Soc Nephrol.* 2015;10(6):923-925

Page 6, line 21: “selection of marginal donors recipients” – not clear what authors are referring to with this phrase

Over the last few decades, the selection criteria has changed to accept more marginal characteristics for donors (e.g. increased age, obesity, diabetes, smoking history) and recipients (e.g. increased age, obesity, previous cancer, health co-morbidities). This is due to to rising numbers for people with organ failure listed for solid organ transplantation and improvements in delivery of care and therapies. This

sentence has been clarified on page 5 to reflect this.

Line 41: Seems like a missed opportunity to forego mentioning that not all cancers have screening tests available;

Also missing, mention of PTLD and its early importance

A comment in relation to both these statements is now made on page 6.

Table 1: how are recipient organs (same sites and combine, such as heart and lung) counted in this table?

This snapshot is based upon a review of raw online data of the UK Transplant registry. Specific details with regards to single or combined organ transplants will be available once the data is linked and made available for analysis.

Table 2: importance of missing race should be discussed perhaps in "Missing Data" section; could another dataset be merged that is more likely to have race/ethnicity available to add to the linkage?

This is mentioned on page 15. It is very likely missing data with regards to ethnicity in the UK Transplant registry will be available in one of the other data registries that will be linked.

Page 11: Nice list of pre-specified questions. Is it possible to look at individual level pharmaceutical data as part of this linkage? If so, looking more closely at type of immunosuppression over time, in a time-dependent survival analysis by transplanted organ, would be important. Is that possible? Strongly suggest adding as a pre-specified question.

A very important question but individual level pharmaceutical data is not available from the UK Transplant Registry (or from any of the other data sources) to be able to adequately tackle this.

Page 13, line 50: will general population data from the cancer registry be frequency matched to transplant recipients by age, gender, cancer diagnosis year?

Yes (also by ethnicity).

Reviewer: 2

Comments to the Author

Thank you for the opportunity to review this study proposal. I commend the authors on a comprehensive proposal. This is an ambitious project drawing together multiple national data streams. For completion I believe it should be acknowledged by the authors that this study is not fully prospective since much of the data was collected prior to study initiation and was not specifically captured for this purpose but for other purposes.

Thank you for these comments. We agree this is not a prospective study per se and have now highlighted this as such.

1. Strengths and limitations section

Would the authors mind explaining why only England and not the UK? perhaps related more to the cancer registry data than the NHS data?

This is correct. While the UK Transplant Registry covers data for all of the United Kingdom, the cancer registry and Hospital Episodes Statistics only includes data for England (devolved countries have their own respective registries for this data). Therefore, we have limited our analysis to patients who live and receive their solid organ transplant in England.

Would the authors mind citing proof of that statement that this is the "The largest national linked dataset of solid organ transplant recipients at risk for developing cancer available internationally"? i.e. the search terms in pubmed etc so that it could be replicated

We have amended this sentence to '*A large national linked dataset of solid organ transplant recipients at risk for developing cancer*'. While some cohorts are described as bigger (e.g., in the US), this is the largest national record dataset with country-wide linkage we are aware of. We have also added another key point that this data is England-specific.

It may be worthwhile mentioning here also that the study is not a formal prospective longitudinal study with defined intervals of outcome ascertainment and the data was not captured specifically for this purpose, rather the investigators are proposing to use existing data.

This has been added as one of the key points.

2. Introduction section

I think it's worth acknowledging that at least some of this excess cancer risk is related to improved overall survival of both recipients and the allografts over time leading to increased time at risk.

This has been added as an additional sentence on page 4 to highlight the increased risk due to cumulative exposure to cancer risks (e.g., immunosuppression being key).

difficult to justify using the word "most publications" here, there are published studies which used similar methodology to the authors proposed design. These may not have been in the UK but nonetheless used similar design.

The sentence has been amended to 'some reports'.

"contributes to inferior post transplant outcomes" I think it would be more appropriate to say "these factors could possibly contribute to inferior..." rather than such a definitive statement since we don't have definitive evidence that this is the case, only suggestions.

This sentence on page 6 has now been edited as suggested.

With regard to big data approach - in this regard what additional or novel parameters are being collected as part of this study that would contribute to such data exploration? Otherwise if there are no new features or parameters being collected, it is unlikely that a big data approach to this data will glean any novel insights.

The anticipated dataset, after linkage between different registries, will contain a large and complex amount of data (both structured and un-structured). From our analytical perspective, our methodological approach will require a careful methodological approach that would be in keeping with big data approaches from a transplant perspective (e.g., Massie AB, Kucirka LM, Segev DL. Big data in organ transplantation: registries and administrative claims. *Am J Transplant.* 2014 Aug;14(8):1723-30).

3. Methods section

national cancer registry "all variables of cancer related interest - what is the list of these variables? could they be listed as a supplementary file?"

These variables from the different data resources are now all attached as a supplementary files 1-3.

The data linkage plan and security are comprehensive.
the pre-specified research questions are appropriate.

Thank you

is the comparison to the general population useful? Other studies have found an elevated risk of certain cancers post transplant, I would imagine a similar pattern may be observed in this study albeit with a more precise estimate of risk given the large number of participants going to be included in this study. One must also acknowledge that screening of potential recipients prior to transplant probably introduces confounding such as PSA screening for prostate cancer etc. Is there a standard UK approach to cancer work up pre-transplant?

One of the key aims of this project is to clarify standardised incidence and mortality ratios in view of the discordant data currently in the literature. As one example, recent work from Canada places skin cancer as the leading cause of post-transplant cancer, while in other data it ranks as one of the least common causes of post-transplant cancer mortality. We think it is important for counselling purposes to be able to give accurate comparisons to the general population.

There is no standard UK approach to cancer work up before transplantation, other than a need to ensure candidates are up to date with national screening programs.

4. compare observed and expected risks of specific cancer types section

This may be difficult, although it may be obvious whether or not a period effect exists, adjusting for improvements in general medical care as well other factors such as ascertainment bias. Has a better appreciation for detection of cancer post transplant occurred over time? I would imagine this is the case at least for skin cancers.

Due to the time span, we will expect some changes to occur over different time periods, but these are difficult to predict in advance but worthy of investigation. It is fair to say there is now greater awareness of cancer after solid organ transplantation, while concern for complications like rejection has diminished over time.

Is race well captured in each of the datasets? Can the authors adequately adjust for race?

Ethnicity is well captured in the UK Transplant Registry, although as shown in Table 2 approximately 22.2% is missing in that registry. However, some of that missing ethnicity data will be contained in the other datasets and we expect the final amount of missing ethnicity data to be much lower and allow adequate adjustment.

The application of a machine learning approach is limited in my opinion. The authors have mentioned to date only standard parameters such as age, sex, year of transplant. it is unlikely that machine learning techniques will produce results different from standard traditional regression approaches in this context unless there is a wide range of other parameters planned on being incorporated.

We agree this is a possibility but after we have an opportunity to explore the vast amount of structured and unstructured data from the different data sources (as now shown in the supplementary file), we will be able to make a firm decision with regards to this. However, the wide range of parameters that we will be accessing (some of which have not been explored to the best of our understanding), may give us an opportunity to explore machine learning approaches if deemed viable.

5. adequate power for study events section - Could the authors explain why so for anus, lip and Kaposi.

We have stated that *'apart from cancer sites of anus, lip and Kaposi sarcoma, we should be able to detect an SIR of 2.0 or less with at least 80% statistical power'*. That is because, according to Collett and colleagues which is the only cancer incidence paper for the UK, these three are the commonest occurring cancers. We have now added a note with regards to this.

I commend the approach to missing data.

Thank you.

The formal involvement of PPI is commendable.

Thank you.

Dissemination plan section

The difficulty with the suggestion that guidelines for post transplant cancer surveillance will develop out of this project is that it may not, even with SIRs and excess risk assessments, unless there are particular inflection points after which incidence increases for different types of cancers.

The planned guidelines are not just for post-transplant surveillance, which may or not be accurately possible with the data that will be generated. We anticipate being able to provide more clarity on specific risk factors within the solid organ transplant cohort to guide post-transplant surveillance.

It is likely that some cancers do have specific inflection points. For example, PTLD may have a bimodal peak in incidence (first within the early years after transplant and the second with longer cumulative exposure). However, this may be different for younger versus older solid organ transplant recipients and justifies the need for detailed analyses with a wealth of information available.

Tables and content are appropriate.

Thank you.

VERSION 2 – REVIEW

REVIEWER	Margaret M. Madeleine, PhD Fred Hutchinson Cancer Research Center United States
REVIEW RETURNED	16-Feb-2021

GENERAL COMMENTS	This manuscript provides a clear description of the protocol for a population-based solid organ transplant (SOT) registry in the UK. Analyses are well thought out, and innovations include use of AI for risk prediction models and plan to examine estimates of morbidity requiring hospitalization. Limitations are well addressed, and appropriate caution around change in maintenance medications to tacrolimus-based regimens by year of transplant is noted as an important potential modifier. Figure, tables, and supplementary information are well chosen, including extensive data dictionaries. How often will linkages will be refreshed? Minor suggestions to improve the manuscript include: * Page 13, line 23-24, Please describe or give a reference for methods to be used to select general population sample;
---

	 * Page 19, line 55, overstatement to say "assess the true impact" suggest provide crucial estimates of the impact or similar * Figure 1 what does pseudo anonymized mean? not described in legend or text Please clarify:  * Page 1, line 27 change all to any? * Page 8, line 24, won't repeat transplant recipients' first transplant be included? Not clear as written * Page 9, line 36, what is the relative completion date for HES data compared to most complete transplant and cancer data? * Page 15, line 23, text "SIRs of at least the following magnitude" should be recast as no magnitudes listed in text, only in table * Page 16, line 15, section 251 referenced but not explained until later in ms, page 17; * Page 16, line 52, PPI forum not explained * Page 18, line 23, reference for HES Analysis Guide
--	---

REVIEWER	Prof Donal Sexton Trinity Health Kidney Centre, School of Medicine, Trinity College Dublin, Ireland.
REVIEW RETURNED	03-Feb-2021

GENERAL COMMENTS	Thank you for the opportunity of reviewing the revised manuscript. I am satisfied with the replies from the investigators.
--

VERSION 2 – AUTHOR RESPONSE

Reviewer: 1

Dr. Margaret Madeleine, University of Washington

Comments to the Author:

This manuscript provides a clear description of the protocol for a population-based solid organ transplant (SOT) registry in the UK. Analyses are well thought out, and innovations include use of AI for risk prediction models and plan to examine estimates of morbidity requiring hospitalization. Limitations are well addressed, and appropriate caution around change in maintenance medications to tacrolimus-based regimens by year of transplant is noted as an important potential modifier. Figure, tables, and supplementary information are well chosen, including extensive data dictionaries. How often will linkages will be refreshed? Minor suggestions to improve the manuscript include:

Thank you for these comments and recommended suggestions to improve our manuscript. There are no current plans for a linkage refresh, but this will be reviewed as the project progresses to determine if the need arises for a data refresh.

* Page 13, line 23-24, Please describe or give a reference for methods to be used to select general population sample;

Thank you. We have added clarification and a new citation [40] for this (page 13 of clean/marked manuscript).

* Page 19, line 55, overstatement to say "assess the true impact" suggest provide crucial estimates of the impact or similar

We have altered this to 'crucial estimates of the impact' (page 19 of manuscript).

* Figure 1 what does pseudo anonymized mean? not described in legend or text

Please clarify:

We have changed to anonymous for clarity as that is what will be received by the study investigators.

* Page 1, line 27 change all to any?

We assume this refers to page 4 of the introduction and have accordingly changed 'all' to 'any' post-transplantation cancers.

* Page 8, line 24, won't repeat transplant recipients' first transplant be included? Not clear as written

We have amended this sentence on page 8 of the clean/marked manuscript. We will only analyse the first receipt of any transplant in the study period and exclude any repeat transplant episodes (to exclude double counting the same subject).

* Page 9, line 36, what is the relative completion date for HES data compared to most complete transplant and cancer data?

This is an excellent question and we simply do not know, especially for more historical data. For example, we know HES data between 1997-2001 is sub-optimal but has better completion from 2001 onwards. We will be including supplementary tables documenting record linkage proportion success (if any patients records could not be linked by NHS Digital), overall data completeness and missing data percentages for variables across all datasets.

*Page 15, line 23, text "SIRs of at least the following magnitude" should be recast as no magnitudes listed in text, only in table

We have amended this sentence on page 15.

* Page 16, line 15, section 251 referenced but not explained until later in ms, page 17;

We have omitted the page 16 mention of the section 251 approval and left the discussion under page 17 as it is more relevant there as it is part of our ethical and legal approval.

* Page 16, line 52, PPI forum not explained

This has been explained a bit further on page 16.

* Page 18, line 23, reference for HES Analysis Guide

This has now been added (citation 41).

Reviewer: 2

Dr. Donal Sexton, Trinity College Dublin

Comments to the Author:

Thank you for the opportunity of reviewing the revised manuscript. I am satisfied with the replies from the investigators.

Thank you for your review and recommended suggestions to improve our manuscript.

VERSION 3 – REVIEW

REVIEWER	Margaret Madeleine, PhD Fred Hutchinson Cancer Research Center Seattle, USA
REVIEW RETURNED	01-Mar-2021
GENERAL COMMENTS	Response to my suggestions are adequate.